# Detection and Verification of a Key Intermediate in an Enantioselective Peptide Catalyzed Acylation Reaction

**DOI:** 10.3390/molecules27196351

**Published:** 2022-09-26

**Authors:** Matthias Brauser, Tim Heymann, Christina Marie Thiele

**Affiliations:** Clemens-Schöpf-Institute for Organic Chemistry and Biochemistry, Department of Chemistry, Technical University of Darmstadt, Alarich-Weiss-Straße 4, 64287 Darmstadt, Germany

**Keywords:** acylation, imidazolium ion, mass spectrometry, NMR spectroscopy, organocatalysis

## Abstract

Until now, the intermediate responsible for the acyl transfer of a highly enantioselective tetrapeptide organocatalyst for the kinetic resolution of trans-cycloalkane-1,2-diols has never been directly observed. It was proposed computationally that a π-methylhistidine moiety is acylated as an intermediate step in the catalytic cycle. In this study we set out to investigate whether we can detect and characterize this key intermediate using NMR-spectroscopy and mass spectrometry. Different mass spectrometric experiments using a nano-ElectroSpray Ionization (ESI) source and tandem MS-techniques allowed the identification of tetrapeptide acylium ions using different acylation reagents. The complexes of *trans*-cyclohexane-1,2-diols with the tetrapeptide were also detected. Additionally, we were able to detect acylated tetrapeptides in solution using NMR-spectroscopy and monitor the acetylation reaction of a *trans*-cyclohexane-1,2-diol. These findings are important steps towards the understanding of this highly enantioselective organocatalyst.

## 1. Introduction

The selective acyl transfer onto alcohols is important for various signaling pathways in nature. Plants use acetyl transferases to synthesize volatile esters during fruit ripening. Mammals on the other hand use it to produce neurotransmitters such as acetylcholine [1,2].

The enzyme-substrate interactions involving intermolecular hydrogen bonds or hydrophobic effects inside the pockets of enzymes inspired chemists to develop catalysts that mimic these functions [3,4,5]. By reducing the rather complex enzyme pockets to smaller peptides Miller et al. introduced oligo-peptides that were able to transfer acyl groups from acetic anhydride onto racemic alcohols, producing enantiomeric excess in the acylated product. This reactivity was attributed to the π-methylhistidine moiety and hydrogen bonds stabilizing the catalysts structure [6,7,8,9,10]. An important step forward was published in 2008 by the Schreiner group. The less flexible tetrapeptide (TP) Boc-l(π-Me)-His^a^Gly-l-Cha-l-Phe-OMe **1** enabled enantiomeric excess values of >99% for the acetylation of racemic *trans*-cycloalkane-1,2-diols (rac)-**2** in toluene (See Figure 1) [11]. In addition to toluene other solvents were tested for this highly enantioselective kinetic resolution of (rac)-**2**. The use of dichloromethane (DCM) and trifluoromethylbenzene resulted in lower conversion and significantly lowered the selectivity, while almost no conversion or selectivity was observed in acetonitrile [11]. Later investigations explored the scope of other electrophiles as well. Using, e.g., isobutyric anhydride **4** as an electrophile the reaction still showed high selectivity and enantiomeric excess values, while for trimethylacetic anhydride **5** almost no conversion and selectivity were observed. This was attributed to steric hindrance [12]. 

The high enantioselectivity was proposed to originate from a pocket-like structure formed by tetrapeptide **1** as a result of incorporating γ-aminoadamantane carboxylic acid into the peptide. This building block increased the stiffness of the peptide backbone while promoting enantiospecific interactions of the *trans*-cycloalkane-1,2-diols (rac)-**2** and the catalytically active π-methylhistidine moiety [11]. Calculated transition states for the acylation of racemic *trans*-cyclohexane-1,2-diols using Density-Functional Theory (DFT) conducted by Shinisha et al. favor the *R*,*R* configured transition state, in accordance with the experiment [13]. The obtained transition state structures resemble the previously proposed pocket-like arrangement of the peptide and the *trans*cyclohexane-1,2-diols. To account for dispersion interactions between catalyst and substrate in the calculations, Müller et al. published a DFT reoptimized pocket-like structure of the acetylated peptide using a functional parametrized for medium range correlations [14]. The ring of the *trans*-cyclohexane-1,2-diol in this model is in close proximity to the cyclohexane moiety of the peptide [12,15]. This predicted interaction could later be correlated with an experimentally observed Nuclear Overhauser Effect (NOE) [16] contact between the cyclohexane moiety of tetrapeptide **1** and (1*R*,2*R*)-cyclohexane-1,2-diol (*R*,*R*)-**2**. In the same work a solution conformer ensemble of the unacylated tetrapeptide **1** was established using NOE and Residual Dipolar Coupling (RDC) data affirming pocket-like conformations to be present in solution for **1** [17]. 

In the mechanism proposed for the reaction of tetrapeptide **1** and the *trans*-cyclohexane-1,2-diols (rac)**-2**, the π-methylhistidine moiety is supposed to transfer the acyl group onto the diols. Thus far, acylated intermediates of tetrapeptide **1** (structures 6–8 in Figure 1) have not yet been observed experimentally. Alachraf et al. were able to detect a related, acylated tetrapeptide using High Resolution ElectroSpray Ionization Mass Spectrometry (ESI-HRMS) [18]. 

We therefore set out to investigate if we can spectroscopically detect different tetrapeptide acylium ions—using three of the previously employed anhydrides (**3**, **4** and **5**)—in toluene and DCM using ESI-HRMS [19,20] and confirm the covalent nature of these intermediates as well as their reactivity using Nuclear Magnetic Resonance (NMR)-spectroscopy. Additionally, the complexation of *trans*-cyclohexane-1,2-diols (rac)-**2** with tetrapeptide **1** and the acetylated intermediate **6** was investigated using nano-ESI-HRMS.

## 2. Results and Discussion

### 2.1. ESI-HRMS Measurements

As a starting point we used a solution of tetrapeptide **1** in DCM and added anhydride **3** in excess to form the corresponding acetylated tetrapeptide **6**. By spraying this solution into a mass spectrometer recording in positive ion mode, a full scan ESI-HRMS spectrum was obtained. The acetylated tetrapeptide **6** was validated using a Higher-Energy Collision Dissociation (HCD) fragment spectrum of precursor ion **6** as shown in Figure 1. (for more details see Appendix A).

After formation of acetylated tetrapeptide **6** was detected and validated, we repeated the experiment with anhydrides **3**, **4**, and **5** in DCM and toluene, successfully detecting the expected masses of the corresponding acylium ions under all conditions with a mass error of smaller 5 ppm (See Table 1). 

The intensities of the cations found follow the experimentally observed trend of reactivity in the acylation of the trans-cyclohexane-1,2-diols: The acetylated tetrapeptide **6** displayed the highest ion count, followed by the isobutyrylated tetrapeptide **7**, and, with very little observed ions, the pivalylated tetrapeptide **8** [12]. Fragmentation of ions selected in MS^2^ experiments was carried out to confirm the identity of the respective ions (See Appendix A for spectra).

After the successful detection of the acylated tetrapeptides **6**, **7**, and **8** the acyl transfer onto *trans*-cyclohexane-1,2-diols was investigated by ESI-HRMS in the negative ion mode. Here, it would be intriguing if the non-covalent interactions between tetrapeptide **1** (or acetylated tetrapeptide **6**) and the diol **2** or the respective acetylated diol **12** could be detected by MS. For this, solutions of tetrapeptide **1** and either *R*,*R*-*trans*-cyclohexane-1,2-diol (*R*,*R*)**-2** or *S*,*S*-*trans*-cyclohexane-1,2-diol (*S*,*S*)**-2** in DCM with 1% acetic anhydride **3** and 1% acetic acid **15** were prepared. DCM was used as a solvent in this case to increase the concentration, since the solubility of tetrapeptide **1** in DCM is higher than in toluene. The addition of acetic acid **15** to the solutions improved the ionization of the molecules (Figure 2).

In this experiment the [M-H]^−^ ion of tetrapeptide **1** (*m*/*z* = 759.4451) is only present in small amounts, the acetate adduct [M+Ac-H]^−^ can be easily detected with high signal to noise ratio at *m*/*z* = 819.4657. Additionally, an ion, the mass of which is in-line with a complex between tetrapeptide **1** and the acetylated diol **12** (called reaction product complex from now on) can be observed by SIM of its precursor mass *m*/*z* = 917.53 in very low abundance. The composition of this ion is validated by matching the fragments of the HCD spectra of tetrapeptide **1** with those of the detected precursor mass of the reaction product complex as shown in Figure 3 (See Appendix A for equivalent experiment using *S*,*S*-*trans*-cyclohexane-1,2-diol (*S*,*S*)**-2**). 

The obtained HRMS fragment masses can be assigned to tetrapeptide **1** as their parent structure (the fragments observed were the same as in toluene for the acetate adduct of **1** with *m*/*z* = 819.4667 (See Appendix A)). This shows that the detected ions of the complex with *m*/*z* = 917.5394 contained tetrapeptide **1**. To further analyze the reaction product complex HCD dissociation curves of the complex at *m*/*z* = 917.53 as well as the adduct complex of tetrapeptide **1** and acetic acid **15** were acquired to allow a relative comparison of their stability (Figure 4).

After a DoseResp [21] fit using Origin (Pro), version number 9.8.0 [22], employing equation Equation (1), the E_50_ value is obtained, where A1 equals the bottom asymptote, A2 the top asymptote, E_50_ the center and p the Hill slope of the curve.
(1)$$y=A1+\frac{A2-A1}{1+10^{(E_{50}-x)p}}$$

The E_50_ value corresponds to the normalized collision energy (NCE) needed to dissociate 50% of the complex and can be used as a measure of the relative stability of the complex [23]. For TP **1** + acetate an E_50_ of 12.5 NCE is obtained while for the reaction product complex an E_50_ of 13.5 NCE is determined. The difference of 1 NCE between the two E_50_ correlates with higher binding affinities between tetrapeptide **1** and acetylated diol **12** in contrast to the acetate adduct of tetrapeptide **1** [24]. Since electrostatics play a large role in the interactions between molecules in ESI-MS measurements [20], the high electron density on the oxygen-atoms of **12** or **2** allows for strong interactions with **1**, e.g., through hydrogen bonding that is preserved over the measurement. We could thus theorize that the delocalized charge of the acetate ion interacts less strongly with **1**. Hence, the higher binding affinity with the already acetylated product **12**, supports the hypothesis that during the acetylation of the diol (*R*,*R*)**-2** a complex is formed either with tetrapeptide **1** or the acetylated tetrapeptide **6** before the acyl residue is transferred onto diol (*R*,*R*)-**2**. Thus, the detection of the acylium ions **6**, **7** and **8**, as well as the complexes of the acetylated diols and tetrapeptide **1** using HRMS was successful.

### 2.2. NMR Measurements

To provide further evidence for the existence and connectivity of the acylated tetrapeptides in solution various NMR experiments were performed on solutions of tetrapeptide **1** and the anhydrides **6**, **7**, and **8** in DCM-*d*_2_ (See Appendix A for more details on sample compositions). First of all, resonances indicative of the formation of the acylated tetrapeptides need to be observed. Considering that there are no previously published NMR spectra of the acylated tetrapeptides and the low ion count observed in the MS measurements, the effective concentration of the acylium ion in solution appears to be low. DCM was again used as a solvent to increase the concentration. Figure 5 shows the changes in the ^1^H-NMR spectrum upon addition of acetic anhydride **3** to tetrapeptide **1** in DCM-*d*_2_.

In addition to several chemical shift changes of tetrapeptide resonances, especially for the amide protons, signals of acetic acid **15** and acetic anhydride **3** become apparent. Further, the highlighted new signal at 10.5–10.6 ppm is notable. A similar signal was found in a solution of isobutyric anhydride **4** and tetrapeptide **1** but no signal in this shift range was observed for trimethylacetic anhydride **5**, congruent with the low reactivity of **5** observed in reactions by the Schreiner group (See Appendix A for spectra) [12]. As the chemical shift is in accordance with protons in similar chemical environments in imidazolium-based ionic liquids in organic solvents, the signal observed is proposed to belong to H29_ac_ [25]. To obtain further evidence of the connectivity in the acylation site the ^13^C-chemical shift of the respective carbon C29_ac_ needs to be obtained. The ^1^H-^13^C-Heteronuclear Single Quantum Coherence (HSQC) [26] spectrum shows a correlation to a carbon chemical shift of 139 ppm for H29_ac_, while carbon C29 of the unacetylated tetrapeptide **1** has a ^13^C-chemical shift of 137 ppm in the same spectrum (See Appendix A for ^1^H-^13^C-HSQC). Thus, carbon C29_ac_ is attributed to the signal with a chemical shift of 139 ppm.

After having obtained the relevant ^13^C chemical shift information, a ^1^H-^13^C-Heteronuclear Multiple Bond Correlation (HMBC) spectrum was recorded, which allows to link spin systems via the coupling between nuclei over more than one bond (Figure 6) [27] (Appendix A).

A correlation of C29_ac_ and a ^1^H signal at 2.7 ppm is observed (blue). This ^1^H signal is attributed to the methyl protons of the acetyl fragments considering its chemical shift, as well as the other correlation the signal is showing (red). This other correlation with a ^13^C resonating at 165 ppm is assigned to be one with the carbonyl carbon of the acetyl fragment. These observed correlations are indicative of a covalent bond between the acetyl fragment and the imidazolium ring and thus prove the existence and successful NMR-spectroscopic characterization of acetylated tetrapeptide **6**. 

By addition of acetic anhydride **3**, the tetrapeptide/DCM-*d*_2_ system re-equilibrates and the ^1^H- as well as the ^13^C-chemical shifts of the imidazole moiety change over time. As the system is observed while the reaction proceeds, the chemical shifts measured differ slightly between the HSQC and the HMBC. 

For more evidence for the connectivity between the two fragments along with new insights into the dynamic process observed during the acylation of the tetrapeptide, selective ^1^H Nuclear Overhauser Enhancement (NOE) spectra [28] were recorded (See Appendix A for experimental details). The resonances of H29, H29_ac_, and Methyl_ac_ were selectively irradiated and their correlations through NOE and chemical exchange are shown in Figure 7.

Evidently, the imidazole proton H29 shows correlations with H29_ac_ and Methyl_ac_. The same interactions are found in the spectra of H29ac and Methyl_ac_ (For spectra in toluene see Appendix A). Since the cross peaks bear the same phase as the diagonal peaks, this indicates that they are the result of exchange between the species. Depending on the correlation time of the compound, this conclusion can be misleading in the case of NOE spectra. It has been previously observed for the tetrapeptide that its correlation time is close to the one that leads to zero crossing of the NOE at 700 MHz [17]. For small molecules (in the case of exchange narrowing [29,30]) NOE (correlations through space) and chemical exchange (then termed EXchange SpectroscopY (EXSY) [31,32]) can be differentiated by their phase. The excited signals and the diagonal signals exhibit the opposite phase for NOE correlations while for chemical exchange they appear with the same phase in the spectrum. For larger molecules (higher molecular weight and thus higher correlation time [33]) a zero crossing for the NOE occurs and in the slow motion limit [34,35] the signal phases in NOE experiments cannot discriminate between NOE and chemical exchange anymore. A remedy to this problem is provided by Rotating-Frame nuclear Overhauser Effect SpectroscopY (ROESY) spectra in which the cross relaxation does not change sign, due to its transversal nature such that spatial information and chemical exchange can be safely discriminated [36]. Efficient Adiabatic SYmmetrized Rotating-Frame nuclear Overhauser Effect SpectroscopY (EASY-ROESY) spectra were recorded for this reason [37]. The EASY-ROESY spectra confirm the exchange between the unacetylated and the acetylated terapeptide species (See Appendix A for EASY-ROESY spectrum) and are thus evidence for tetrapeptide **1** being in equilibrium with the acetylated tetrapeptide **6** during the NMR experiment. 

After detection of the different acylated tetrapeptides and verification of their structural connectivity, the next step is to actually follow the acetylation of the diol (Figure 1). Thus, a solution of tetrapeptide **1** and acetic anhydride **3** in DCM-*d*_2_ was prepared. The high solubility of **1** in DCM-*d*_2_, combined with the fact that the acetylated tetrapeptide investigated here, shows the most intense signal of the acylated tetrapeptides, allows the study of its reaction with *R*,*R*-*trans*-cyclohexane-1,2-diol (*R*,*R*)**-2** using NMR-spectroscopy. A time series of 1D-^1^H-spectra was recorded as a pseudo 2D-spectrum directly after adding (*R*,*R*)**-2** to a solution of **1** and **3**. Signals of the *R*,*R*-*trans*-cyclohexane-1,2-diol (*R*,*R*)**-2** (grey) vanish, while the signals of the acetylated diol **12** (green, blue and red) build up (Figure 8 (left)), resulting in the reaction profile shown in Figure 8 (right) (See Appendix A for sample composition). The signal intensity of the acetylated tetrapeptide **6** was constant over the course of the reaction. This reaction profile can be fit using DynaFit [38]. The best correlation with a low root-mean-square deviation (RMSD) of 0.21 mmol/L is obtained for a first order process. This suggests a direct acetylation of (*R*,*R*)**-2** by the acetylated tetrapeptide **6** under the conditions chosen (pseudo first order since acetic anhydride **3** is in large excess as ratio **3**:**1** = 12.20 and ratio (*R*,*R*)**-2**:**1** = 1.20) (See Appendix A for first order fit and Appendix A for zeroth- and second order fits for comparison).

## 3. Conclusions

We investigated the acyl-transfer by a highly selective tetrapeptide organocatalyst **1** onto *trans*-cyclohexane-diols. We employed different mass spectrometry techniques for the detection of acylated and unacylated species of tetrapeptide **1** alone and in complex with (acetylated) *trans*-cyclohexane-diols. The acylium ion of tetrapeptide **1** could be detected for all three anhydrides chosen. NMR spectroscopy was used to confirm the proposed acetylation site. The connectivity between the acyl fragment and peptide organocatalyst **1** could unambiguously be determined by HMBC. The acetylation takes place at the π-methylhistidine moiety as proposed previously. NMR was further used to monitor the reaction of the prepared catalyst. Under the conditions chosen (acetic anhydride **3** in excess) the signal of the acylium ion **6** could be observed throughout the reaction and its intensity remained constant. The detection of tetrapeptide **6** and the reaction product complexes is an important step towards the understanding of the highly enantioselective reaction of this organocatalyst.

## Data Availability

The data presented in this study are available on request from the authors.

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
