# Peer review of "Detection and Verification of a Key Intermediate in an Enantioselective Peptide Catalyzed Acylation Reaction"

_molecules, 2022, doi:10.3390/molecules27196351_

Round 1
Reviewer 1 Report
In this communication, Brauser M. et al report a detailed mechanistic and analytical investigation into the critical acylated imidazolium ion intermediate in enantioselective peptide catalyzed acylation of trans-1,2-cyclohexanediol. While the details of the mechanism and intermediate have been determined computationally, this intermediate has not been observed experimentally. The authors use sophisticated NMR and MS techniques to support the presence of this intermediate. The authors have significant expertise in these areas and the methods are more than adequately performed. While the MS and NMR results clearly support the presence of the intermediate as well as the acylation being in equilibrium, this intermediate has already been universally accepted as the most probable pathway, and it is difficult to image alternatives. The results do no necessarily shed new light on this reaction. Nevertheless, experimental evidence for this intermediate is important to have. I only have a couple comments for the authors to consider.
-From the HCD dissociation curves, the E50 value is greater for the reaction product complex suggesting greater binding, however the authors do no posit an explanation. The alkoxide in the product anion has a more localized charge distribution which will enable it to engage in stronger H-bonding with the peptide than the acetate anion. Perhaps the authors could elaborate on this or a related point.
-In the last sentence of the conclusions, the authors mention this is an important step in understanding the enantioselectivity, however the spectroscopic techniques used were not able to probe any of the complex interactions that govern the enantioselectivity. Perhaps the authors could mention that detection of the intermediate and products is important for understanding this enantioselective reaction, but not necessarily the reasoning for the high enantioselectivity.
Author Response
- We thank the reviewer for his appreciation !
- Concerning the comment on the HCD curves: Since we did not want to speculate about the charge localization of the reaction product, we only showed the most probable one. Because of the very helpful comment of reviewer 1 we were able to elaborate more on this and mention it in the captions. As electrostatic interactions in ESI-MS measurements are important and influence the binding affinity of the compounds, we mentioned this effect at the respective passage too.
- Concerning enantioselectivity: We changed the last sentence of our conclusion to emphasize that our study is an important step in understanding the overall reaction and not primarily the enantioselectivity of this reaction. We noticed that the same was true for the abstract and also changed that one.
Reviewer 2 Report
The authors reported detailed studies on catching one of the key intermediate in acylation. The evidence showed in this report i believe is enough to deliver the conclusion. And the result is helpful for scientists working on this chemistry to understand the origin of selectivity. This paper can be accepted with minor revision on the format of references, for example, capitalize all initial words of Ref.14,19,20,21....
Author Response
We capitalized all initial words in the references according to the original publication (if the original publication document was capitalized differently than the capitalization used on the website of the publisher). For the sake of consistency we updated the capitalization of the manuscript’s title.
Reviewer 3 Report
Authors describe a full spectrosocipc anamysis (MS and NMR mainly) to assess a mechanism of acylation using a tetrapeptide.
The work is very interesting and deserves to be published after minor modifications:
Abstract should be a summary of the work and not a pre-introduction. Just focus on results
In scheme 1, if compounds 6-8 are intermediates, those compounds should be drawn between brackets
"acylated intermediates of tetrapeptide 1 have not yet been observed experimentally." Maybe those species should be drawn.
Author Response
- We have shortened the abstract accordingly.
- The notation of how to draw intermediates is not consistent in literature; but usually brackets are used for transient – non-isolatable species. As we have spectroscopically characterized these intermediates – one of them even in solution in NMR spectroscopy – this means that they are present for an extended period of time. Thus; the notation with brackets that would be chosen for really transient species did not seem appropriate to us. Thus we did not change anything here.
- Maybe this is a misunderstanding. These compounds (6-8) were already drawn in the Figure from the beginning. To make the connection between the mentioned acetylated intermediates of tetrapeptide 1 clearer we added a reference to scheme 1 at the respective position.